# CombiANT reader: Deep learning-based automatic image processing tool to robustly quantify antibiotic interactions

Erik Hallström[1]*, Nikos Fatsis-Kavalopoulos[2], Manos Bimpis[2], Carolina Wählby[1,3], Anders Hast[1], Dan I. Andersson[2]

**1** Department of Information Technology, Uppsala University, Uppsala, Sweden, **2** Department of Medical Biochemistry and Microbiology, Uppsala University, Uppsala, Sweden, **3** SciLifeLab Science for Life Laboratory, Uppsala University, Sweden

* erik.hallstrom@it.uu.se

**Data availability statement:** The authors confirm that all data underlying the findings are

## Abstract

Antibiotic resistance is a severe danger to human health, and combination therapy with several antibiotics has emerged as a viable treatment option for multi-resistant strains. CombiANT is a recently developed agar plate-based assay where three reservoirs on the bottom of the plate create a diffusion landscape of three antibiotics that allows testing of the efficiency of antibiotic combinations. This test, however, requires manually assigning nine reference points to each plate, which can be prone to errors, especially when plates need to be graded in large batches and by different users. In this study, an automated deep learning-based image processing method is presented that can accurately segment bacterial growth and measure distances between key points on the CombiANT assay at sub-millimeter precision. The software was tested on 100 plates using photos captured by three different users with their mobile phone cameras, comparing the automated analysis with the human scoring. The result indicates significant agreement between the users and the software ($0.7 \pm 0.39$ mm mean absolute error) and remains consistent when applied to different photos of the same assay despite varying photo qualities and lighting conditions. The speed and robustness of the automated analysis could streamline clinical workflows and make it easier to tailor treatment to specific infections. It could also aid large-scale antibiotic research by quickly processing hundreds of experiments in batch, obtaining better data, and ultimately supporting the development of better treatment strategies. The software can easily be integrated into a potential smartphone application, making it accessible in resource-limited environments. Integrating deep learning-based smartphone image analysis with simple agar-based tests like CombiANT could unlock powerful tools for combating antibiotic resistance.

fully available without restriction. A replication package is available at https://doi.org/10.5281/zenodo.13893288 containing all image data and software to reproduce the experiments, generate output metrics and build the graphs in this article.

**Funding:** Funding was awarded to D.A. and C.W. from the Swedish Foundation for Strategic Research, https://strategiska.se/en, grant number: SSF ARC19-0016. The computations were enabled by the Berzelius supercomputing resource provided by the National Supercomputer Centre at Linköping University and supported by the Knut and Alice Wallenberg Foundation. Additionally, computations were enabled by the Alvis cluster provided by the National Academic Infrastructure for Supercomputing in Sweden (NAISS) at Chalmers University of Technology, partially funded by the Swedish Research Council through grant agreement no. 2022-06725. The funders had no role in study design, data collection and analysis, decision to publish, or preparation of the manuscript.

**Competing interests:** The authors have declared that no competing interests exist.

## Author summary

Antibiotic resistance is a significant problem worldwide with a high prevalence of multi-resistant bacteria that may require the simultaneous administration of several different antibiotics. With the right antibiotics and concentrations, such combination therapy may treat a strain that is otherwise resistant to each antibiotic individually. CombiANT is a novel test that can identify suitable or inappropriate antibiotic combinations. However, it requires the human evaluator to grade each plate manually, which is time-consuming, and errors can easily be made, especially if the human evaluator needs to grade many plates in succession. In this study, an image processing pipeline is developed using a deep neural network to grade CombiANT test assays automatically.

## 1. Introduction

Antibiotics are a cornerstone of modern healthcare, allowing treatment of bacterial infections that were once fatal or severely disabling. However, their effectiveness is increasingly threatened by antibiotic resistance, driven by antibiotic overuse and misuse [1]. One way to combat the resistance is by prescribing combination therapies. Some antibiotics can work together in synergy, resulting in a greater efficacy than when each one acts alone. Conversely, some antibiotics may reduce each other's effects, a phenomenon referred to as antagonistic effects [2–6]. This interaction is also dependent on the respective concentration of each antibiotic. Therefore, it is of high importance to carefully evaluate and understand the interactions between different antibiotics before prescribing combination therapies.

One novel method to assess antibiotic interactions is CombiANT, an easy-to-use assay that allows for the testing of antibiotic synergies on agar plates [7]. The CombiANT test involves an agar plate containing three 3D-printed reservoirs, each accommodating a different antibiotic. Following preparation, a diffusion landscape forms, generating different concentrations of each antibiotic at every location on the plate. The assay is shown in Fig 1 with the inserts containing antibiotics marked with A, B, and C. There are two growth zones, "outer" and "inner", shown as darker areas with uninhibited bacterial growth. The white triangle is the interaction area where antibiotics act in pairs given by distance to the inner growth zone, dashed lines AB, AC, and BC. The outer distances, dashed lines A, B, and C, indicate the efficacy of the antibiotic acting alone. In the original CombiANT software, a user has to manually annotate nine points in the correct order, outlined with red numbers 1–9 in Fig 1. A detailed explanation of CombiANT is outlined in S1 Appendix.

Up until the 2010s, the field of computer vision and computerized image processing mainly consisted of manipulating images using classical methods such as various filters [8,9], edge detection [10], thresholding [11], and segmentation algorithms such as the watershed transform [12]. Following the winning of AlexNet [13] in the Large-scale image recognition challenge in 2012, it became evident that training deep artificial neural networks with convolutional filters (CNN) could significantly outperform other classical methods in various image processing tasks, including biomedical segmentation as demonstrated by the ground-breaking U-Net architecture [14]. Furthermore, in 2020, the Vision Transformer (ViT) [15], a deep artificial neural network with a different architecture, containing no convolutional filters, was shown to be on par with or outperforms convolutional neural networks.

At present, no computer vision application for automatically scoring the CombiANT assay is available, even though some related image processing methods exist, including the automated counting of colony-forming units (CFU) on an agar plate [16]. This task relates to the

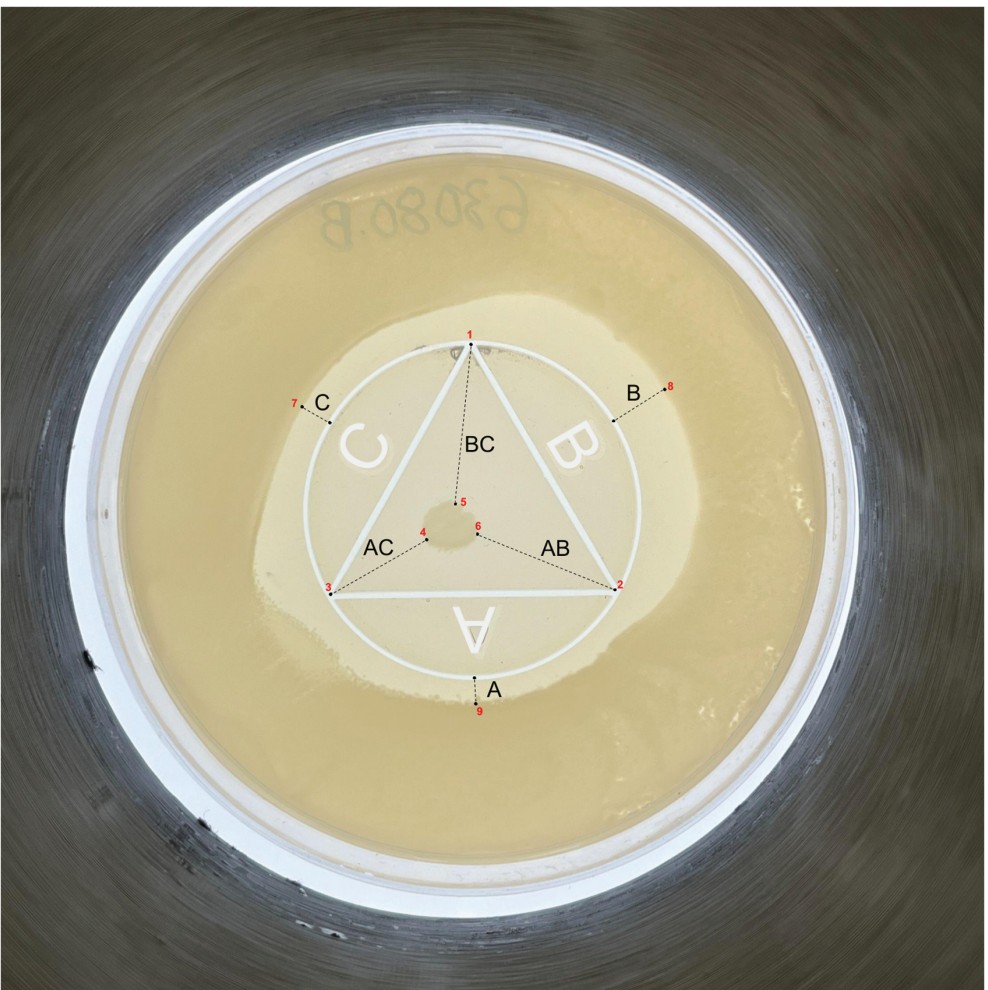

**Fig 1. CombiANT.** A CombiANT assay with nine points that has to be manually pinpointed by a human evaluator. Dashed lines show distances indicating the effects of antibiotics acting alone (A, B, and C) or in pairs (AB, AC, BC) due to the proximity of the reservoirs. The three antibiotic inserts (reservoirs), A, B, and C, are outlined in white for visibility.

growth-zone segmentation in the CombiANT assay in the proposed pipeline, as both require identifying and segmenting bacterial content on the agar plate while discarding non-bacterial objects and artifacts. Several image processing methods have been proposed to accomplish this, utilizing both classical [17–23] and deep learning-based [24–27] approaches. Another related image processing task is measuring the inhibition zone in the Kirby–Bauer disk diffusion test. This zone is typically circular, hence, the Hough Circle transform [28] is commonly used to measure the inhibition [29,30], although various deep-learning models have been explored [31]. In the CombiANT test, the inhibition zone is not circular, therefore this prior cannot be used.

To address the lack of automated analysis of antibiotic interactions, an image processing pipeline was developed in this study for this specific application. The deep learning method (U-Net) was selected for the main task of the pipeline to segment bacterial growth zones, due

to the proven robustness and superior performance of CNNs demonstrated in recent years across various image processing tasks. This is exemplified by the mentioned related colony counting works utilizing classical methods, which often require good imaging conditions with even illumination without shadows or glare and a high-end camera (typically achieved by photographing the plates inside a box with a vertically mounted camera [17,21,23]). The following highlights how our work differs from the aforementioned related colony counting works utilizing deep learning: Shamash et. al [26] and Nagy et al. [27] pose the task as an object detection problem (using the YOLOv5 model [32] and Faster R-CNN [33], respectively) which is not applicable in our case, since we need to obtain the shape of the growth zone. Ferrari et al. [25] extract colony aggregate images using a proprietary segmentation procedure built into the Copan WASPLab automation system and then use a custom-designed CNN to count the colonies, posing it as a classification problem. Hence, their method cannot be used for our task. Among the related works, only Andreni et al. [24] share some similarities with our proposed solution, using a fully convolutional Pyramid Scene Parsing Network [34] to segment colonies. However, this network is more tailored for semantic segmentation of natural images with global context and employs more aggressive downsampling with ResNet [35] backbone. The U-Net was selected for higher spatial precision, which is required when segmenting bacterial growth zone boundaries. Furthermore, our pipeline is required to perform additional tasks not present in the related works, finding the inscribed triangle and measuring the closest distance to the boundary of the growth zones.

The overall objective of the study is to demonstrate the speed, robustness, and efficiency of using deep learning-based image analysis to automatically grade agar plate-based antibiotic tests such as the CombiANT. By eliminating manual annotation, this approach has the potential to significantly impact antibiotic resistance research by facilitating large-scale studies, improving reproducibility, and ultimately contributing to more effective treatment strategies.

## 2. Results

The proposed CombiANT Reader software method automates the annotation process described in the original CombiANT test. The software finds the growth zones and triangle vertices and measures the required distances at sub-millimeter precision, visualized in Fig 2. The main advantage of this automated approach compared to the existing CombiANT methodology is reliability, as currently the human evaluator has to annotate the points on the plate (shown in Fig 1) in the correct order. Additionally, using the software could lead to significant potential time savings and experiment scalability benefits.

The main component of the developed image processing pipeline is the U-Net [14], a deep fully convolutional neural network used to segment growth regions of bacterial content in the assay. However, the software also utilizes classical image processing methods, such as locating the center triangle using template matching in OpenCV [36] (see Materials and Methods for details).

The developed software was evaluated on 100 CombiANT assays, having three different users taking a picture of each plate and then independently evaluating the plates using the current CombiANT software. The users were labeled as "Beginner" (User 3), "Intermediate" (User 1), and "Experienced" (User 2). The beginner user performed the first-ever grading of plates, while the experienced user had previously performed many gradings. As described in the Introduction section, this involved manually annotating points on the growth zone boundaries and triangle mark vertices, seen in Fig 1. Then, the developed image processing

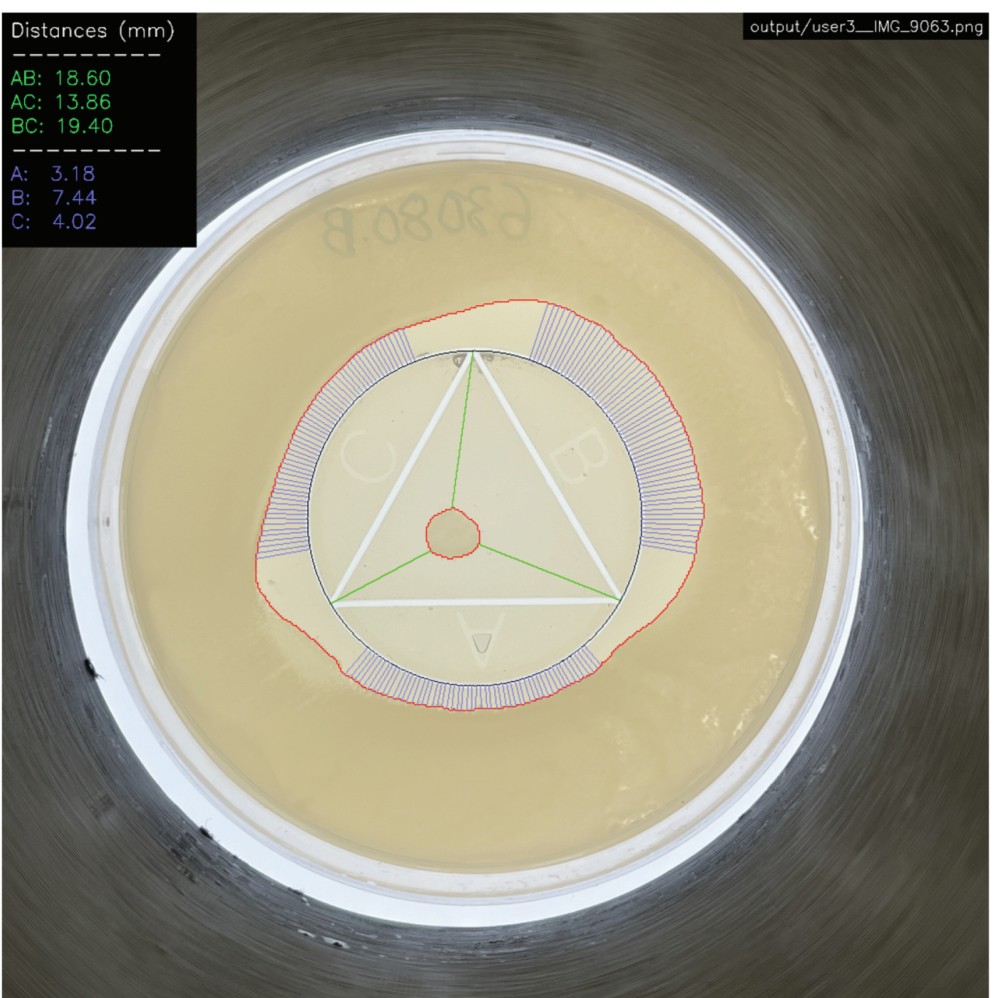

**Fig 2. CombiANT reader.** The CombiANT Reader software automatically finds the circle mark with the inscribed triangle, outlined with a black border, and the edges of the growth zones, outlined in red. The software measures 55 distances from each reservoir to the outer growth zone (blue lines) and automatically finds the closest distance to the inner growth zone from every triangle vertex (green lines). The legend shows the calculated distances in millimeters with two decimals. For the outer distances, the legend shows the median value for the respective reservoir.

software was tested by processing images from all users (300 images in total), one sample shown in Fig 2. The software automatically finds the inner and outer bacteria growth zones, the white triangle-in-circle mark, and the closest distance from each triangle vertex to the inner growth zone (AB, AC, and BC). The software measured a number of distances from the circle perimeter to the outer growth zone (55 by default, this can be tuned). The median of these distances was calculated for each well and used in our analysis (A, B, and C). Software-annotated outputs from six plates from all three users are visualized in S1–S18 Figs. The results show high agreement between the manually scored assays and the users despite different lighting conditions and varying distances from the plate, results are shown in Table 1.

**Table 1. Mean Absolute Error (mm) between user and software gradings, breakdown on user and distance.**

| Distance | Beginner | Intermediate | Experienced | Distance MAE |
|---|---|---|---|---|
| A | 0.49 ± 0.27 | 0.47 ± 0.26 | 0.42 ± 0.29 | 0.46 ± 0.27 |
| B | 0.47 ± 0.82 | 0.40 ± 0.35 | 0.38 ± 0.36 | 0.41 ± 0.51 |
| C | 0.42 ± 0.35 | 0.50 ± 0.30 | 0.49 ± 0.44 | 0.47 ± 0.36 |
| AB | 0.82 ± 0.55 | 0.82 ± 0.26 | 0.77 ± 0.52 | 0.80 ± 0.44 |
| AC | 0.91 ± 0.54 | 0.95 ± 0.36 | 0.66 ± 0.33 | 0.84 ± 0.41 |
| BC | 1.25 ± 0.38 | 1.13 ± 0.36 | 1.29 ± 0.33 | 1.22 ± 0.35 |
| User MAE | 0.73 ± 0.48 | 0.71 ± 0.32 | 0.67 ± 0.38 | |

## 2.1. Outer distances

The outer distances A, B, and C, as annotated by the users, were compared to the software grading. This task was relatively more straightforward than measuring the inside distances since the latter required identifying the point in the inner growth zone closest to the corresponding triangle vertex. As shown in Figs 3, 4, and 5, the software aligns with the user gradings and is more consistent having a lower standard deviation for each measurement.

## 2.2. Inner distances

Estimating the inner distances (AB, AC, and BC) was more challenging for the users due to the difficulty in identifying the point closest to the triangle vertex. As seen in Figs 6, 7, and 8 the software often estimates a larger distance than the users.

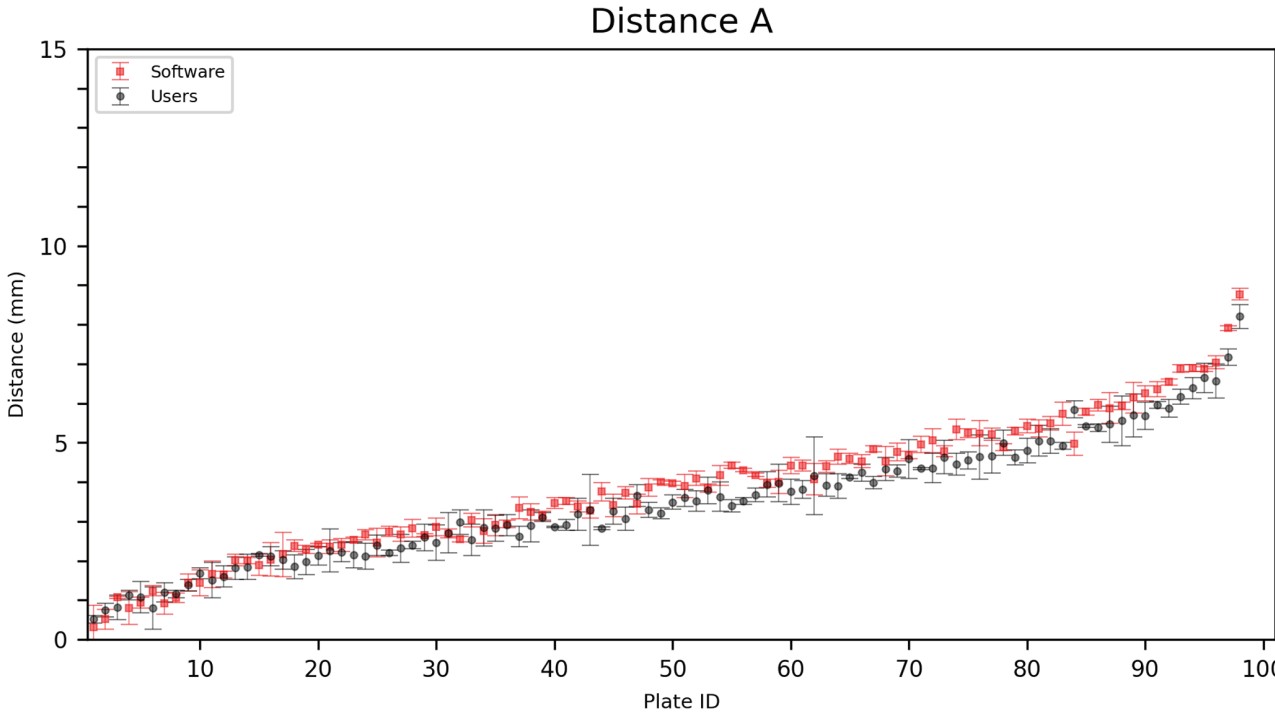

**Fig 3. Comparison of user and software gradings for Distance A.** The plates are arranged on the horizontal axis in ascending order according to the mean of the measured distances (combining software and user distances). Error bars show the mean and standard deviation of the three user and software measurements, respectively.

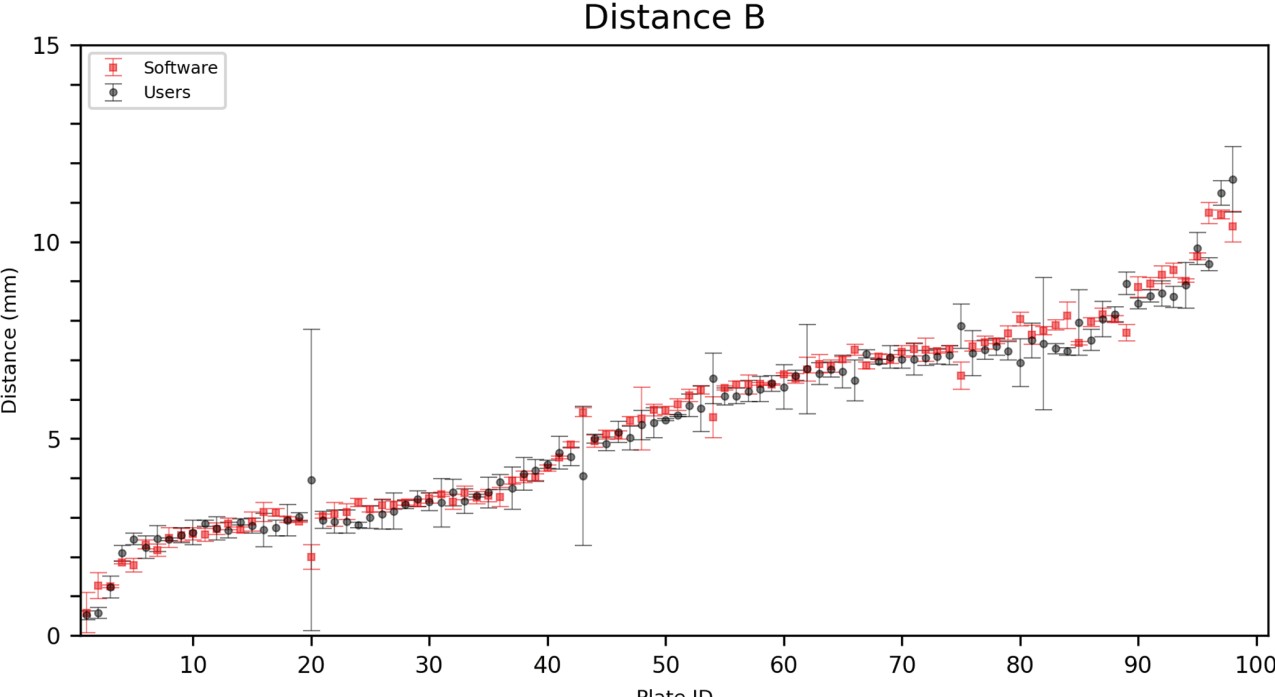

**Fig 4. Comparison of user and software gradings for Distance B.** The plates are arranged on the horizontal axis in ascending order according to the mean of the measured distances (combining software and user distances). Error bars show the mean and standard deviation of the three user and software measurements, respectively.

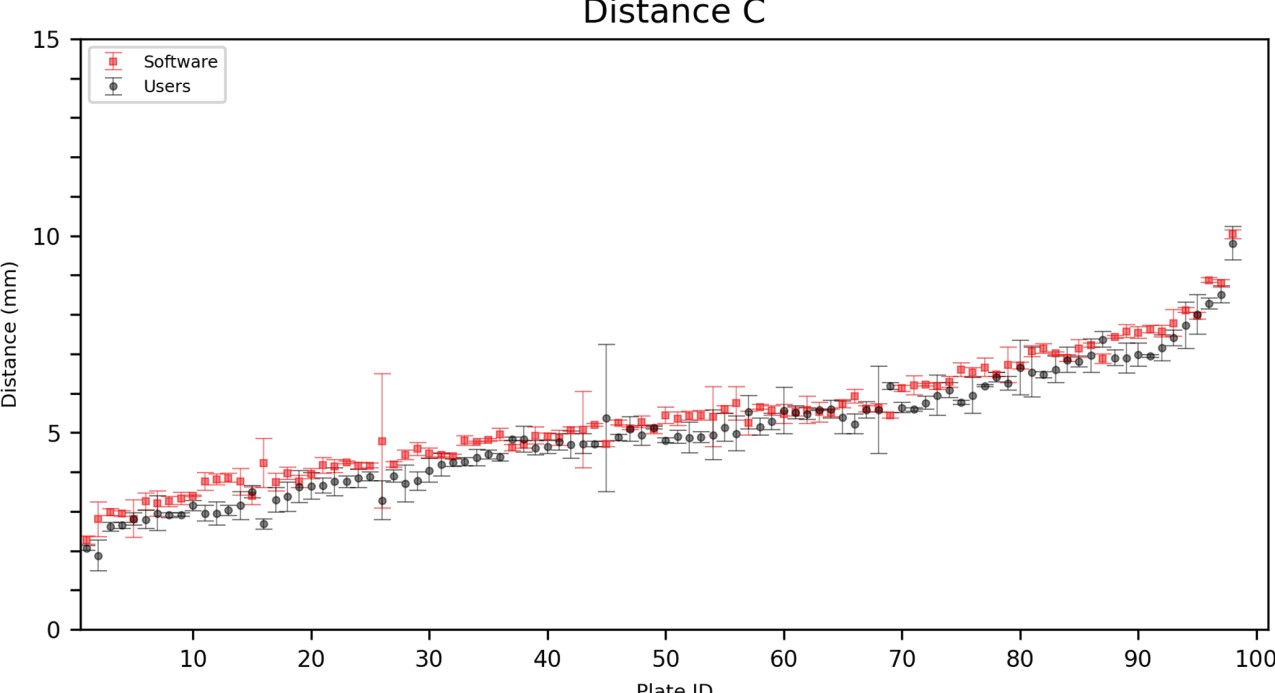

**Fig 5. Comparison of user and software gradings for Distance C.** The plates are arranged on the horizontal axis in ascending order according to the mean of the measured distances (combining software and user distances). Error bars show the mean and standard deviation of the three user and software measurements, respectively.

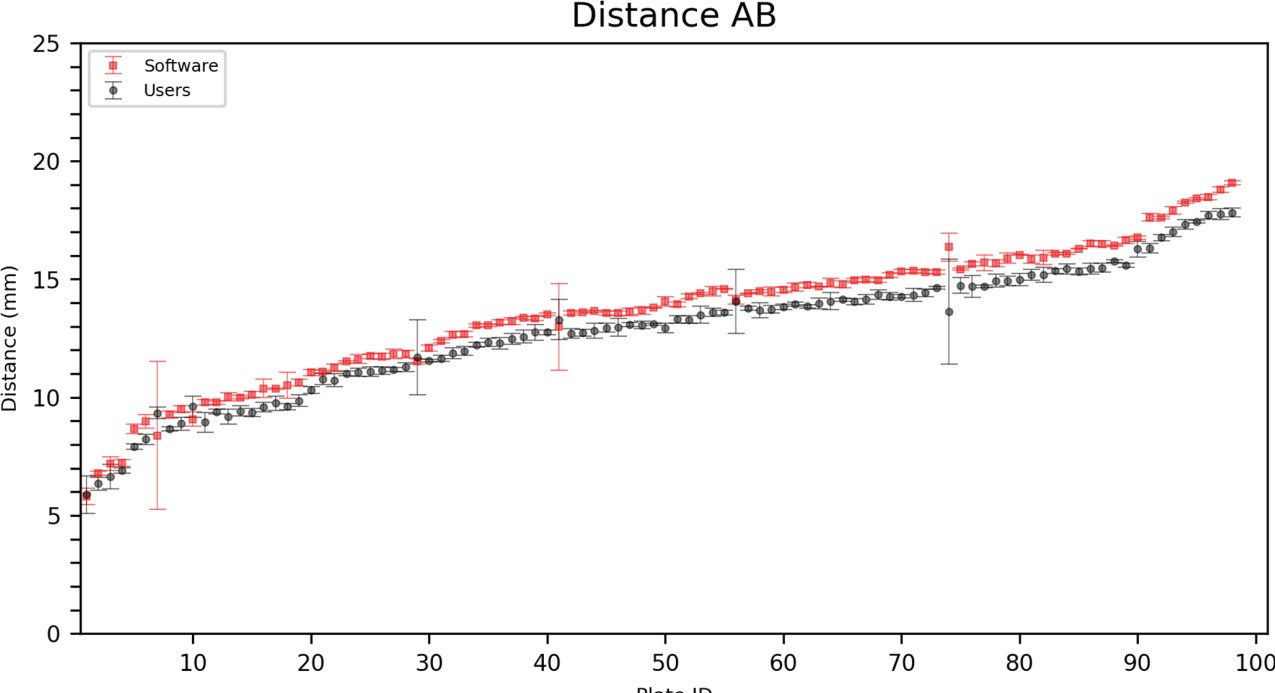

**Fig 6. Comparison of user and software gradings for Distance AB.** The plates are arranged on the horizontal axis in ascending order according to the mean of the measured distances (combining software and user distances). Error bars show the mean and standard deviation of the three user and software measurements, respectively.

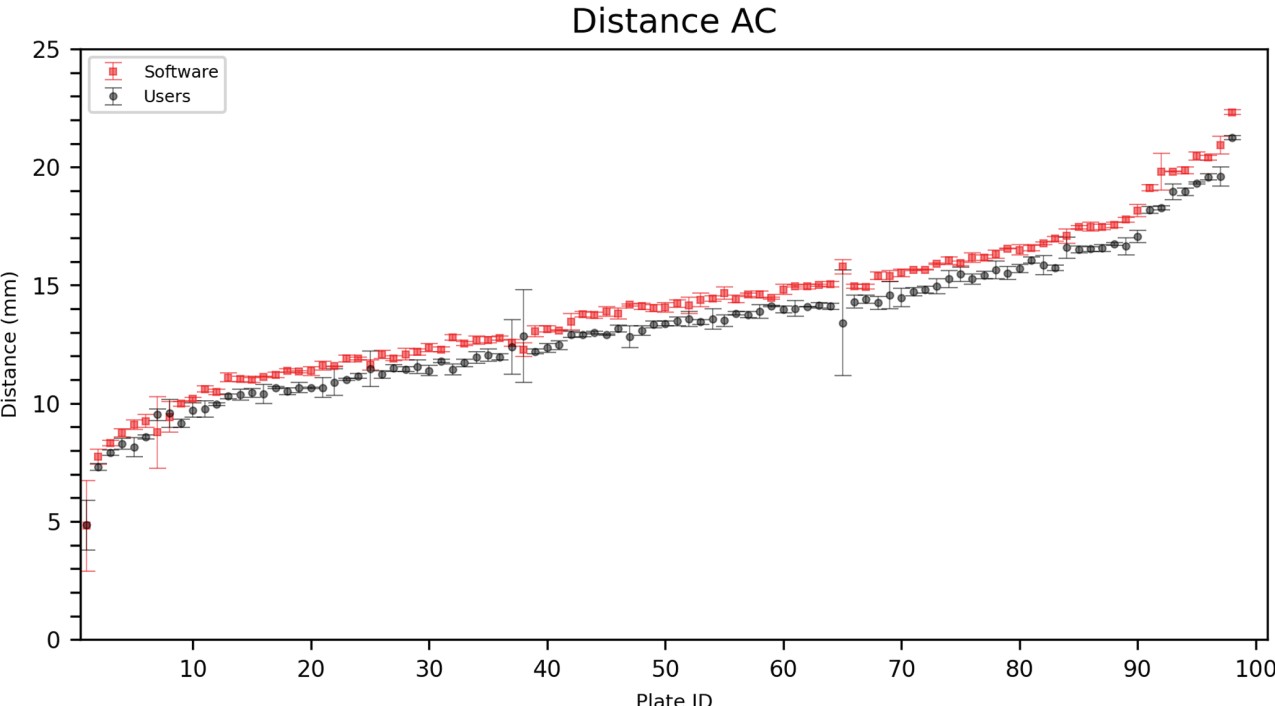

**Fig 7. Comparison of user and software gradings for Distance AC.** The plates are arranged on the horizontal axis in ascending order according to the mean of the measured distances (combining software and user distances). Error bars show the mean and standard deviation of the three user and software measurements, respectively.

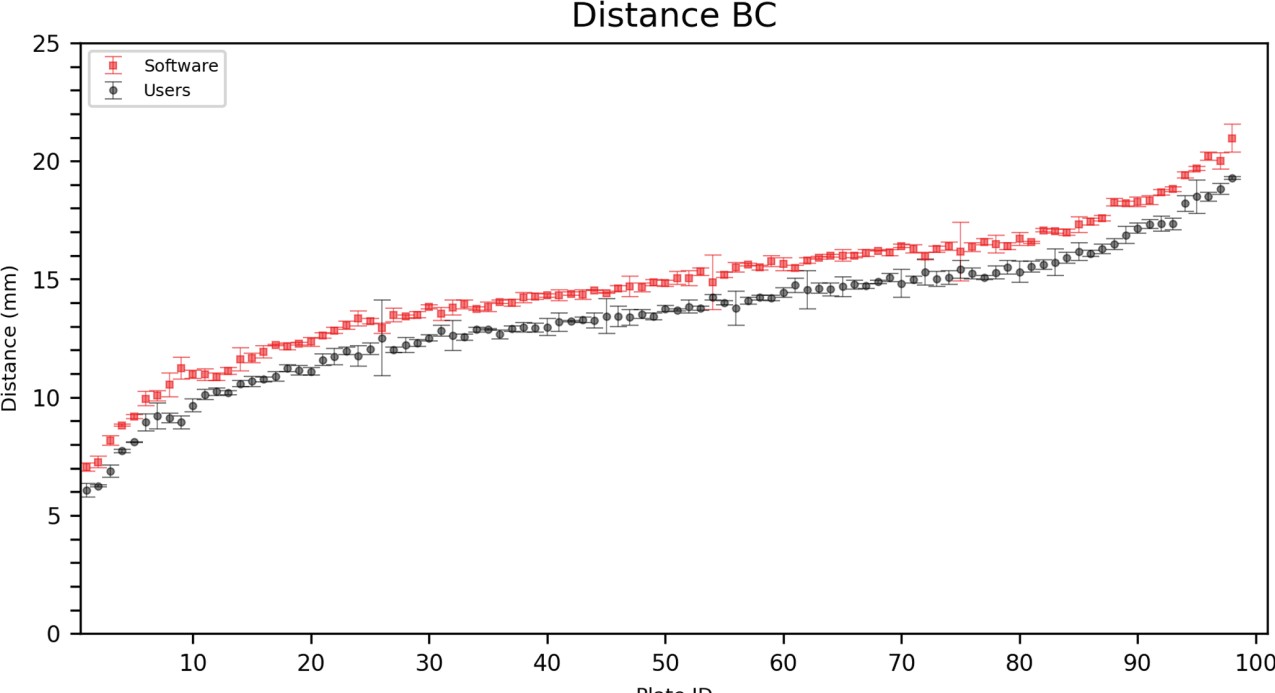

**Fig 8. Comparison of user and software gradings for Distance BC.** The plates are arranged on the horizontal axis in ascending order according to the mean of the measured distances (combining software and user distances). Error bars show the mean and standard deviation of the three user and software measurements, respectively.

In order to investigate the overestimation, the absolute difference between the software and user grading was calculated for each image and inner distance. As shown in Fig 9, the smallest differences were observed with the experienced user, suggesting that the developed software grading aligns more closely with the experienced user's evaluations.

## 2.3. Discarding of plates and outliers

The developed pipeline includes a feature to discard plates where the assay is deemed unreliable, for example, if the growth zone overlaps the interaction zone or if no inner growth zone is detected, exemplified in S19–S20 Figs (see Discarding of plates in Materials and Methods for details).In the tests, the human evaluators and the software always agreed on whether to discard a plate for all test plates. However, these plates were still included in our evaluation as there were distances that could still be used. All discarded plates are shown in S2 Appendix.

In total, five distances from four plates contained an absolute error between user and software gradings above 3.5 mm. These outliers are visualized in S3 Appendix. One outlier was an error by the software, and the rest by the beginner user.

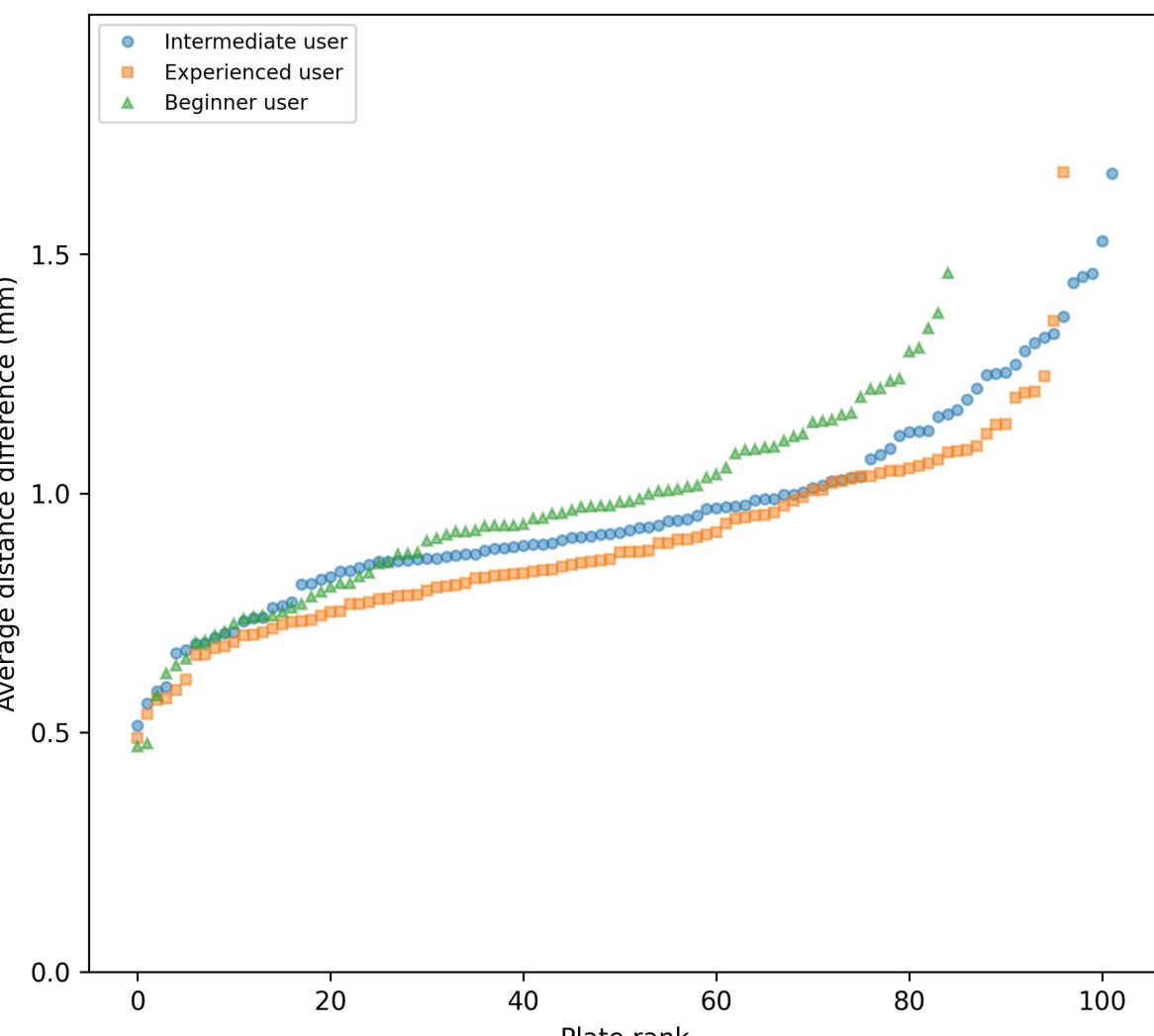

**Fig 9. Comparison of the difference between user and software gradings for the inner distances.** For each image, the absolute difference between the user and software grading of the inner distances was calculated and then averaged over the three inner distances. The plates are sorted on the horizontal axis independently for each user according to the difference, in ascending order. Three outliers, one from the experienced user and two from the beginner user are not visible in the plot.

## 3. Discussion

This paper presents an image processing pipeline for automatically grading CombiANT assays. The pipeline can process photographs from smartphone cameras, and the software can either be implemented on the client in a smartphone application, or the analysis could be performed on the server side in the cloud. The pipeline is consistent with human evaluation (0.7 $\pm$ 0.39 mm MAE), robust to different imaging conditions, and fast, taking only a few seconds per plate. If scaled to larger clinical research settings, the processing speed is unlikely to be the bottleneck, the current software can process 12000 plates per day on a modern workstation computer (see "Inference time" in Materials and Methods). Furthermore, the processing time can be greatly improved, especially the template matching step.

The growth zone segmentation does not require any particular imaging setup or
7 illumination, which is required by many of the classical methods [17–19]. Using heavy augmentation during training, the model performs well even when challenged with imaging conditions not present in the training set. If the segmentation fails, more data can easily be annotated, followed by retraining the network. In contrast, if the classical segmentation methods were used, a correction would have meant tuning existing parameters or introducing a composition of new filters and operations, which may have successfully managed to segment the failed sample but instead infer an error in another image in the training set.

Several optimizations can be made to the software in future work. Most importantly, the template matching part could be replaced with an end-to-end deep learning model that automatically finds key points (keypoint estimation), is robust to rotations shear and (if the agar plates were photographed at a tilted angle), and compensates for perspective effects. Such models could also read the marked reservoir letters on the assay (A, B, and C) and not require the CB vertex to be pointed upwards, a disadvantage of the current solution. One approach to accomplish this could be to annotate the triangle vertices in each image and generate heatmaps with Gaussian blobs centered on them. The U-Net would then output four channels: one for bacterial content and three for the triangle's vertices. At inference, the argmax of the three vertex channels will be computed to locate the vertices. Then the image will be rotated so that AB points upwards, followed by an affine transformation to align the points to an equilateral triangle.

Plenty of architectural improvements have been made since the inception of U-Net in 2014. For example, the Vision Transformer (ViT) [15] has been adapted for image segmentation, as seen in Segformer [37] and other developments. Segformer uses a ViT backbone and thus has better global context but is optimized for natural scene segmentation. A drawback is that it relies on token-based representations, which can result in less precise boundaries and lower granularity in segmentation. We believe that for the problem of bacteria growth zone segmentation, U-Net or its variants are sufficient and remain highly effective.

We hope the developed software will inspire future work in image analysis software for processing other easy-to-use assays similar to CombiANT that have yet to be envisioned and could be analyzed using smartphone photos in low-resource settings.

Furthermore, most agar-based combination tests that produce segmentable inhibition zones—such as the Double Disk Synergy Test (DDST), Combination Disk Test (CDT), and D-Test—could adopt the methodology presented in this paper. Additionally, checkerboard tests using microtiter plates could be analyzed by image processing software that first locates the wells, followed by assessing the content based color intensity, turbidity or other optical properties. In a research setting, a potential approach could involve a robot preparing thousands of plates, which are then analyzed by deep learning image processing systems. Such innovations will both reduce the workload and increase the robustness of the assay when assessing interactions between antibiotics, thereby facilitating work in clinical microbiology laboratories as well as in research settings.

## 4. Materials and methods

Here, the main components of the developed pipeline are described. However, for full implementation details, refer to the released software package [38], which contains documentation, a pre-trained U-Net segmentation model, annotated training data, and the test images to re-run the evaluation. The software was developed using Python 3.9.19, apart from the libraries mentioned below, the software also utilized the NumPy [39] and Pandas [40] libraries.

All images were captured using standard smartphone cameras, positioning the CombiANT assay on a flat surface and taking the photo from directly above. The raw images had a size of around 3000x3000 pixels. The image processing pipeline was written using the OpenCV [36] image processing library and PyTorch [41] and Albumentations [42] for deep neural network training. Initially, using bicubic interpolation, the software made a square center crop and downscaled the plate images to 1024x1024 pixels.

## 4.1. Segmentation of bacterial growth zones using U-Net

The pipeline employed U-Net [14], a fully convolutional artificial neural network, for segmenting the bacterial growth zones. The network was trained on segmentation masks from 1000 separate CombiANT plates not present among the plates in the evaluation. The masks were constructed by manually highlighting the bacteria boundaries using standard image processing software and then converting these images to segmentation masks. Initially, a subset of the training set was annotated, this data was then used to train a network to annotate the remaining unannotated images. The instances where the network made errors were then corrected. This strategy enabled the quick annotation of all images in the training set.

The network was trained for 1000 epochs using batch size 20, learning rate $10^{-4}$, Cosine learning rate scheduler, and the ADAM [43] optimizer. The hyperparameters were chosen based on typical standard values, and the ADAM optimizer is less sensitive to the base learning rate as each parameter has an individually adapted learning rate. The batch size was set to 20 (instead of 32) to fit 512×512 pixel images in GPU memory, and the number of epochs was increased due to heavy augmentation.

The training time was 8 hours using the Nvidia A100 GPU. Upon receiving images from the test users to perform the evaluation, it was observed that the test images significantly differed from the training images, showing variances in background, illumination, and contrast. This caused errors in the segmentation, so the networks were retrained, adding additional augmentations, after which the model successfully segmented most of the test images. However, the contrast was manually increased using histogram equalization on a few of the evaluation images for the segmentation network to function (see Manual preprocessing).

The output segmentation mask was then inverted, and only one connected component with the largest area was filtered out, resembling the empty space without bacterial content in the center of the agar plate (the inhibition zone). The mask was then inverted again, effectively removing all holes in the segmentation mask. This process was followed by discarding connected components with a minimum area, only allowing two bacterial growth zones, "outer" and "inner".

The segmentation was performed by first subsampling the raw images to 512x512 pixels using the "inter_area" method in OpenCV, followed by U-Net inference and thresholding, and then subsequently upsampling the output mask back to 1024x1024 pixels.

## 4.2. Alignment using template matching and grid search

Each assay was marked with a white equilateral triangle inscribed into a circle, as seen in Figs 1 and 2. This is referred to as the "triangle-in-circle"-mark or just "mark." The corners of the triangle and the position and radius of the center circle of the mark were retrieved using a procedure utilizing template matching from the OpenCV library. First, a square center crop was extracted from the plate containing the mark, followed by applying a circular binary mask at the center, effectively zeroing out pixels outside the circle-mark perimeter.

Next, the colored image was converted to a grayscale image and thresholded at the 98th percentile, obtaining a binary image revealing the bright pixels of the image, which coincided with the mark. A function was constructed, returning a binary template image of an equilateral triangle inscribed into a circle, resembling the mark, with parameters adjusting the scale (size), rotation, and line thickness. A number of these templates were generated and matched to the thresholded image utilizing an adaptive grid search outlined below.

### 4.3. Adaptive grid search

An adaptive grid search was used to obtain the optimal parameters of the template that best match the thresholded image containing the mark. The procedure first performed a search using scale, then rotation. The reason for starting with scale is that the correct scale, the circle part of the mark, overlaps with the template regardless of orientation. The procedure searched for a number of values above and below a start value with a predetermined step size (0.01 and 1 for rotation and scale, respectively), obtaining the best fit, location, and value (scale or rotation). The fit and location were obtained from the OpenCV template matching algorithm. The grid search algorithm used is outlined in Algorithm 1.

**Algorithm 1: Adaptive grid search for optimal template matching parameters.**

```
Require: I Thresholded image containing the triangle in circle mark
Require: F Function to compute the template matching score for a given scale or
    rotation of the binary template image.
Require: step_size The step size for the grid search
Require: start_value The start value for the grid search
Require: n_steps The number of steps to search in each direction
Require: n_levels The number refining search levels
Require: min_fit Expand the search if fit below this value
Require: max_tries Maximum tries to expand the search
Output: best_value The best scale or rotation of the template
```

1: $best\_score \leftarrow -1$
2: $best\_value \leftarrow -1$
3: **for** $lvl \leftarrow 1$ to $n\_levels$ **do**                    ▷ Iterate over refinement levels
4:     **for** $try \leftarrow 1$ to $max\_tries$ **do**   ▷ Expand the search in the first level if necessary
5:         **for** $step \leftarrow -n\_steps$ to $n\_steps$ **do**                         ▷ Grid search
6:             $value \leftarrow start\_value + step \times step\_size$
7:             $score \leftarrow F(value, I)$
8:             **if** $score > best\_score$ **then**
9:                 $best\_score \leftarrow score$
10:                $best\_value \leftarrow value$
11:             **end if**
12:         **end for**
13:         **if** $best\_score > min\_fit$ **or**
14:         $abs(best\_value) \neq n\_steps \times step\_size$ **or**
15:         $lvl \neq 1$ **then**
16:             $start\_value \leftarrow best\_value$                    ▷ Update start value for the next level
17:             $step\_size \leftarrow step\_size/n\_steps$                         ▷ Refine the step size
18:             **break**
19:         **end if**
20:     **end for**
21: **end for**

**4.3.1. Expanding search.** If the value with the best fit was the minimum or maximum value in the grid search, the search was retried with the number of search values doubled,

using the identical step size. The search was also expanded if the best fit was below a predetermined threshold.

**4.3.2. Refining search.** Once the approximate best-fit value was identified, a finer search was performed around that optimal value with a smaller step size. This refining search was executed once by default but could be expanded with more levels. The step size was adjusted to search around the best-fit value, extending from the previous value to the next value (relative to the best-fit on the previous level).

Following this grid search, the triangle-in-circle mark was outlined on each test sample with a black border, as seen in Fig 2 and S1–S18 Figs.

## 4.4. Measurement of distances

After the triangle vertices and position of the circle-mark perimeter were found using the template matching step, distances were measured on the assay. First, the binary mask from the U-Net processing, where connected components resembled bacteria content, was converted to contours using the OpenCV "drawContours"-function. The measurements were performed by drawing straight lines on the binary mask containing the contours from key points of interest until the line encountered a contour. The end point of the line, and thus the distances, was obtained by finding the point where the contour and line intersected.

**4.4.1. Inner growth zone.** The minimum distance along a straight line from each triangle vertex to the inner growth zone was measured and visualized as green lines in Fig 2 and S1–S18 Figs. The distances are shown as AB, AC, and BC in the legend, corresponding to the respective triangle vertex. The measurement was obtained by drawing a number of test lines from the corresponding triangle vertex to the inner growth zone. The line with the shortest distance was retained as the minimum distance.

**4.4.2. Outer growth zone.** A total of 55 distances were measured per well, drawing lines perpendicular to the circle-mark perimeter to the rim of the outer growth zone. A 20-degree padding was added on each side of every triangle-mark vertex since the combination concentration is larger there (two antibiotic wells intersect). The outer distances are supposed to measure susceptibility to one antibiotic only. The median of the distances per well was calculated and shown as A, B, and C in the figure legend in Fig 2 and S1–S18 Figs.

**4.4.3. Millimeter conversion.** All distance measurements were initially obtained in pixels. Then, the resolution $R(i)$ was calculated for image $i \in test\_images$ using the radius of the circle-mark known from the fabrication ($r_{mm} = 20.5mm$), and the radius of the current image $r(i)_{px}$ in pixels, obtained from the template matching step.

$$R(i) = r(i)_{px}/r_{mm}$$

The resolution for the images was at 10.069 +- 1.604 px/mm (mean, standard deviation). Using this information, all pixel distances $d_{px}$ were converted to millimeters $d_{mm}$ for all images $i$, $d_{mm} = d_{px}/R(i)$. If the plates were photographed from a far distance, there could be a precision issue, but this was not a problem in practice.

## 4.5. Manual preprocessing

One of the users (User 3) did not align the plates before photographing, pointing the CB vertex up. We manually cropped and rotated these images. U-net segmentation failed on five images from User 1 and 16 images from User 3. We manually increased the contrast of these images, converting to YCrCb color space, and increased the contrast in the Y-channel (which corresponds to the brightness) using histogram equalization.

### 4.6. Discarding of plates

A plate was discarded when any of three criteria were met:

- The inner growth zone grew outside of the triangle-mark interaction zone.
- The outer growth zone crossed the circle mark perimeter at a point where outer distances were measured (not considering points within the 20-degree padding from each triangle vertex).
- There was no center growth zone, it disappeared due to the high synergetic effect of the antibiotics.

These conditions were detected in the pipeline with added discard warnings, shown as "inner", "outer", and "no_island" on the legend. These plates were still processed and distances measured, but with the warning attached, see S18–S20 Figs.

### 4.7. Inference time

Processing one sample took around 7 seconds, with U-net inference taking $0.0376 \pm 0.0283$ seconds ($1.039 \pm 0.446.$ seconds when running on the CPU) and template matching and grid search taking $7.240 \pm 1.660$ seconds.

### 4.8. Use of artificial intelligence tools and technologies

The tools Grammarly and ChatGPT were used for grammar checking, as a synonym book, and as rephrasing tools when writing this article. No original content was generated by the models. Furthermore, the coding assistant GitHub Copilot was used to generate type annotations and docstrings for the software in the released replication package.

## Supporting information

**S1 Appendix. CombiANT.** A detailed explanation of the CombiANT assay and its annotation process.
(PDF)

**S2 Appendix. Discarded plates.** All seven plates flagged for discarding are visualized in one document, along with the rationale for why the plates were discarded.
(PDF)
**S3 Appendix. Outliers.** All four plates where user and software measurements differed by more than 3.5 mm are visualized in one document, along with a possible explanation for the outliers.
(PDF7)

### Output images

Six plates from the study are shown below, photographed by each of the three users and annotated with an overlay created by our CombiANT reader. Our developed software finds the circle mark with the inscribed triangle using template matching, outlined with a black border, and the edges of the bacterial growth zones, outlined in red. The software measures 55 distances from each reservoir to the outer growth zone (blue lines) and automatically finds the closest distance to the inner growth zone from every triangle vertex (green lines). The legend

shows the calculated distances in millimeters with two decimals. For the outer distances, the legend shows the median value for the respective reservoir.

**S1 Fig. Plate 1 photographed by User 1.**
(PNG)

**S2 Fig. Plate 1 photographed by User 2.**
(PNG)

**S3 Fig. Plate 1 photographed by User 3.**
(PNG)

**S4 Fig. Plate 2 photographed by User 1.**
(PNG)

**S5 Fig. Plate 2 photographed by User 2.**
(PNG)

**S6 Fig. Plate 2 photographed by User 3.**
(PNG)

**S7 Fig. Plate 3 photographed by User 1.**
(PNG)

**S8 Fig. Plate 3 photographed by User 2.**
(PNG)

**S9 Fig. Plate 3 photographed by User 3.**
(PNG)

**S10 Fig. Plate 4 photographed by User 1.**
(PNG)

**S11 Fig. Plate 4 photographed by User 2.**
(PNG)

**S12 Fig. Plate 4 photographed by User 3.**
(PNG)

**S13 Fig. Plate 5 photographed by User 1.**
(PNG)

**S14 Fig. Plate 5 photographed by User 2.**
(PNG)

**S15 Fig. Plate 5 photographed by User 3.**
(PNG)

**S16 Fig. Plate 6 photographed by User 1.**
(PNG)

**S17 Fig. Plate 6 photographed by User 2.**
(PNG)

**S18 Fig. Plate 6 photographed by User 3.**
(PNG)

### 4.9. Examples of discarded plates

**S19 Fig. Inner discard** Discarded plate due to inner growth zone intersecting with the triangle-mark.
(PNG)

**S20 Fig. Outer discard** Discarded plate due to the outer growth zone growing past the circle mark.
(PNG)

## Author contributions

**Conceptualization:** Erik Hallström.

**Data curation:** Erik Hallström, Nikos Fatsis-Kavalopoulos, Manos Bimpis.

**Formal analysis:** Erik Hallström.

**Funding acquisition:** Carolina Wählby, Dan I. Andersson.

**Investigation:** Erik Hallström, Nikos Fatsis-Kavalopoulos.

**Methodology:** Erik Hallström.

**Resources:** Nikos Fatsis-Kavalopoulos, Dan I. Andersson.

**Software:** Erik Hallström.

**Supervision:** Anders Hast.

**Validation:** Erik Hallström.

**Visualization:** Erik Hallström, Nikos Fatsis-Kavalopoulos.

**Writing – original draft:** Erik Hallström.

**Writing – review & editing:** Carolina Wählby, Anders Hast, Dan I. Andersson.

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
