## [Decision Letter · Decision Letter 0]

10 Feb 2025

PDIG-D-24-00449CombiANT Reader - Deep learning-based automatic image processing and measurement of distances to robustly quantify antibiotic interactionsPLOS Digital Health Dear Dr. Hallström, Thank you for submitting your manuscript to PLOS Digital Health. After careful consideration, we feel that it has merit but does not fully meet PLOS Digital Health's publication criteria as it currently stands. Therefore, we invite you to submit a revised version of the manuscript that addresses the points raised during the review process. Please submit your revised manuscript within 30 days Mar 12 2025 11:59PM. If you will need more time than this to complete your revisions, please reply to this message or contact the journal office at digitalhealth@plos.org. Please include the following items when submitting your revised manuscript:* A rebuttal letter that responds to each point raised by the editor and reviewer(s). You should upload this letter as a separate file labeled 'Response to Reviewers'. This file does not need to include responses to any formatting updates and technical items listed in the 'Journal Requirements' section below.* A marked-up copy of your manuscript that highlights changes made to the original version. You should upload this as a separate file labeled 'Revised Manuscript with Track Changes'.* An unmarked version of your revised paper without tracked changes. You should upload this as a separate file labeled 'Manuscript'. If you would like to make changes to your financial disclosure, competing interests statement, or data availability statement, please make these updates within the submission form at the time of resubmission. Guidelines for resubmitting your figure files are available below the reviewer comments at the end of this letter. We look forward to receiving your revised manuscript. Kind regards, Ismini LourentzouSection EditorPLOS Digital Health Ismini LourentzouSection EditorPLOS Digital Health Leo Anthony CeliEditor-in-ChiefPLOS Digital Healthorcid.org/0000-0001-6712-6626  **Journal Requirements:**1. We ask that a manuscript source file is provided at Revision. Please upload your manuscript file as a .doc, .docx, .rtf or .tex. **Additional Editor Comments (if provided):****Reviewers' Comments:** Reviewer's Responses to Questions

**Comments to the Author**

1. Does this manuscript meet PLOS Digital Health’s publication criteria? Is the manuscript technically sound, and do the data support the conclusions? The manuscript must describe methodologically and ethically rigorous research with conclusions that are appropriately drawn based on the data presented.

Reviewer #1: Yes

Reviewer #2: Yes

Reviewer #3: Yes

2. Has the statistical analysis been performed appropriately and rigorously?

Reviewer #1: Yes

Reviewer #2: N/A

Reviewer #3: I don't know

3. Have the authors made all data underlying the findings in their manuscript fully available (please refer to the Data Availability Statement at the start of the manuscript PDF file)?

Reviewer #1: Yes

Reviewer #2: Yes

Reviewer #3: Yes

4. Is the manuscript presented in an intelligible fashion and written in standard English?

Reviewer #1: Yes

Reviewer #2: Yes

Reviewer #3: Yes

5. Review Comments to the Author

Reviewer #1: Review for CombiANT Reader - Deep learning-based automatic image processing and measurement of distances to robustly quantify antibiotic interactions

The title:

- The title of the article, "CombiANT Reader - Deep learning-based automatic image processing and measurement of distances to robustly quantify antibiotic interactions," effectively conveys the critical focus and novelty of the study, which lies in utilizing deep learning for automated image analysis in the context of antibiotic interactions. However, I would recommend another version of the title for more precision and clarity, such as : “CombiANT Reader: A Deep Learning-Based Tool for Automatic Image Processing and Quantification of Antibiotic Interactions Using Distance Measurement”

The abstract:

- The abstract is well formed with an apparent problem statement, the relevance of the research question under study, with novel methods, and showed good validation and consistency; however, I have some recommendations that may enhance it; these are:

o Quantify Performance: Include specific metrics to highlight the software's accuracy and reliability compared to human scoring.

o Expand Practical Implications: Briefly discuss how the method could impact clinical workflows or large-scale research on antibiotic resistance.

o Clarify Technical Details: Concisely explain "150 distances" to enhance comprehension.

o Refine Language: Avoid repetitive phrasing and improve the flow for readability.

The introduction:

- Although it has strong points as follow:

o The introduction clearly outlines the global threat of antibiotic resistance and highlights the importance of evaluating antibiotic interactions, particularly for combination therapies. This sets a strong foundation for the study.

o The detailed explanation of the CombiANT assay, including its setup, diffusion dynamics, and manual scoring process, provides valuable context for understanding the study's focus.

o The introduction identifies a significant gap in the lack of automated scoring methods for the CombiANT assay, effectively justifying the development of the proposed deep learning-based solution.

o By referencing related methods such as the Kirby–Bauer disk diffusion test and colony-forming unit (CFU) counting, the introduction situates the study within the broader context of antibiotic testing methodologies.

- I think some weaknesses could be strengthened for better readability and clear ideas, such as:

o Complexity and Redundancy: The introduction is dense with technical details, which may overwhelm readers unfamiliar with the CombiANT assay. Specific descriptions, such as the manual scoring process and triangle vertex annotations, are overly detailed for an introduction and could be summarized.

o Limited Focus on Study Objectives: While the introduction provides ample background, the study’s objectives and potential impact are not emphasized. A stronger focus on how this pipeline advances antibiotic resistance research would strengthen the narrative.

o Lack of Transition to Deep Learning: The introduction transitions abruptly from explaining the manual CombiANT test to discussing the deep learning-based pipeline. A smoother connection highlighting why deep learning is a suitable solution would improve coherence.

o Insufficient Highlighting of Novelty: Although the study’s novelty is implied, it is not explicitly stated. Emphasizing what distinguishes this work from existing methods (e.g., precision, scalability, or adaptability to non-circular inhibition zones) would make the contribution more straightforward.

My Recommendations for the results section are:

o Include Quantitative Validation: Add specific quantitative metrics such as correlation coefficients, mean absolute error, or Bland-Altman analysis to strengthen the argument for the software's accuracy.

o Clarify Outliers and Plate Discarding: Expand on the reasons for outliers and provide examples of the criteria used for plate discarding to ensure transparency.

o Discuss Intermediate User Performance: Highlight the intermediate user’s performance to understand better how the software performs across varying skill levels.

o Expand on Software Advantages: While reliability and consistency are emphasized, the results should also discuss the software's potential time savings and scalability benefits compared to manual scoring.

My suggestions and comments for the discussion section: Although the discussion section has lots of information and is well formed in a clear and informative way, these are my comments that may enhance it and make it more precise and more profound:

o It lacks Quantitative Performance Metrics: The discussion refers to the software's "consistency with human evaluation" but lacks concrete metrics to substantiate this claim. Including statistical results (e.g., accuracy, precision, recall) would make the argument more robust.

o It has Limited Exploration of Scalability: While the discussion mentions cloud-based analysis, it does not elaborate on how the software can be scaled for high-throughput settings, which could enhance its applicability for larger clinical or research facilities.

o It includes a Superficial Description of Future Optimizations: The potential use of transformer architectures and other advanced models is not explored in depth. A brief explanation of how these advancements could address specific challenges (e.g., rotation, perspective effects) would add value.

o It has an Overgeneralization of Future Impact: The statement about inspiring future work is broad. Adding concrete examples or potential areas where the software could be extended (e.g., analyzing other types of agar-based assays or antimicrobial resistance studies) would make this vision more compelling.

My Recommendations for the Materials and Methods section:

o Justify Parameter Choices: Include brief explanations for key parameter selections and how they were optimized during development to add credibility to the methodology.

o Enhance Training Data Description: Provide more details on the diversity of the training dataset (e.g., types of bacteria, variations in assay design) to help readers assess the model's generalizability.

o Automate Preprocessing: Highlight future plans to automate manual preprocessing steps using deep learning models or image normalization techniques, such as contrast adjustment or rotation.

o Discuss Scalability: Expand on how the system could be scaled for high-throughput applications, including computational requirements for cloud deployment.

o Analyze User Bias: Conduct a systematic analysis of user bias by evaluating the impact of user expertise and image quality on segmentation and distance measurements.

Weaknesses and Suggestions for The References section:

1. Inconsistent Citation Style:

o There are inconsistencies in formatting, such as missing capitalization of proper nouns (e.g., "nature" instead of "Nature") and inconsistent punctuation (e.g., missing periods after journal abbreviations).

o Recommendation: Revise the references to adhere to a consistent citation style, depending on the journal's requirements.

2. Over-reliance on General Sources:

o Some references, such as general reviews on deep learning [23], might be less directly relevant to the specific methods used in the study.

o Recommendation: Prioritize references that directly relate to the methodologies or concepts applied in this study, such as segmentation models or antibiotic resistance testing.

3. Limited Context for Certain Citations:

o References like [4] and [5], which discuss specific antibiotic combinations, are poorly integrated into the study’s methodology or results discussion.

o Recommendation: Ensure that these references are explicitly linked to the rationale or implications of the study to enhance coherence.

4. Lack of Historical Context:

o The references focus heavily on recent advances but could benefit from citing foundational works in bacterial resistance and image processing.

o Recommendation: Include seminal works in both fields to provide historical context and highlight how the study builds upon existing knowledge.

5. Absence of Validation-Specific Citations:

o While the study emphasizes validation of the developed pipeline, there are limited references to methodologies for validating deep learning models or image analysis systems.

o Recommendation: Incorporate references to established validation protocols or metrics to strengthen the methodological rigor.

Reviewer #2: It is funning that CombiANT is be used to identify suitable or inappropriate antibiotic combinations to deal with multidrug-resistant bacteria. If there is a another set of validation data, the results will be more credible.

Reviewer #3: This is a useful example of a deep learning digital tool and improves on the current method of quantifying antibiotic interactions. Please add some text on how results were validated, and agreement between 3 reviewers was assessed.

---

## [Decision Letter · Decision Letter 1]

23 May 2025

CombiANT Reader: Deep learning-based automatic image processing tool to robustly quantify antibiotic interactions

PDIG-D-24-00449R1

Dear Mr Hallström,

We are pleased to inform you that your manuscript 'CombiANT Reader: Deep learning-based automatic image processing tool to robustly quantify antibiotic interactions' has been provisionally accepted for publication in PLOS Digital Health.

Best regards,

Ismini Lourentzou

Section Editor

PLOS Digital Health

**Additional Editor Comments (if provided):**

Given the reviewers' valuable effort in providing fine-grained detailed feedback, it is strongly recommended to address any remaining suggested edits in the camera-ready.

**Reviewer Comments (if any, and for reference):**

Reviewer's Responses to Questions

**Comments to the Author**

1. If the authors have adequately addressed your comments raised in a previous round of review and you feel that this manuscript is now acceptable for publication, you may indicate that here to bypass the “Comments to the Author” section, enter your conflict of interest statement in the “Confidential to Editor” section, and submit your "Accept" recommendation.

Reviewer #1: (No Response)

2. Does this manuscript meet PLOS Digital Health’s publication criteria? Is the manuscript technically sound, and do the data support the conclusions? The manuscript must describe methodologically and ethically rigorous research with conclusions that are appropriately drawn based on the data presented.

Reviewer #1: Yes

3. Has the statistical analysis been performed appropriately and rigorously?

Reviewer #1: Yes

4. Have the authors made all data underlying the findings in their manuscript fully available (please refer to the Data Availability Statement at the start of the manuscript PDF file)?

Reviewer #1: Yes

5. Is the manuscript presented in an intelligible fashion and written in standard English?

Reviewer #1: Yes

6. Review Comments to the Author

Reviewer #1: 1. Title

• Partially Addressed: The current title is clear but could benefit from specificity (e.g., replacing "tool" with "method" and explicitly mentioning "distance measurement").

2. Abstract

Suggestions:

• Clarify "150 distances": Not addressed. The term "55 distances" is used without explanation.

• Refine Language: Partially addressed. Phrases like "robustness" and "consistency" are repeated.

3. Introduction

• Simplify Details: Overly detailed descriptions of the CombiANT assay and manual scoring remain.

• Study Objectives: Implicit but not emphasised (e.g., "eliminating manual annotation").

• Transition to Deep Learning: An Abrupt shift from manual methods to U-Net without justifying why deep learning was chosen.

• Novelty: Implied but not explicitly stated (e.g., no direct comparison to existing tools).

4. Results

• Software Advantages: Mentioned ("time savings" in Discussion but not in Results).

5. Discussion

• Performance Metrics: Limited to MAE; lacks precision/recall.

• Future Optimisations: Superficial (e.g., "replace template matching with keypoint estimation").

6. Materials & Methods

• Preprocessing Automation: Manual steps remain (e.g., contrast adjustment).

7. References

• Relevance: Most are relevant.

7. PLOS authors have the option to publish the peer review history of their article (what does this mean?). If published, this will include your full peer review and any attached files.

**Do you want your identity to be public for this peer review?** For information about this choice, including consent withdrawal, please see our Privacy Policy.

Reviewer #1: None
